# Digging into the NGS Information from a Large-Scale South European Population with Metastatic/Unresectable Pancreatic Ductal Adenocarcinoma: A Real-World Genomic Depiction

**DOI:** 10.3390/cancers16010002

**Published:** 2023-12-19

**Authors:** Dimitrios C. Ziogas, Eirini Papadopoulou, Helen Gogas, Stratigoula Sakellariou, Evangellos Felekouras, Charalampos Theocharopoulos, Dimitra T. Stefanou, Maria Theochari, Ioannis Boukovinas, Dimitris Matthaios, Anna Koumarianou, Eleni Zairi, Michalis Liontos, Konstantinos Koutsoukos, Vasiliki Metaxa-Mariatou, George Kapetsis, Angeliki Meintani, Georgios N. Tsaousis, George Nasioulas

**Affiliations:** 1First Department of Internal Medicine, Laikon General Hospital, School of Medicine, National Kapodistrian University of Athens, 11527 Athens, Greece; helgogas@gmail.com (H.G.); hartheoch@gmail.com (C.T.); dimitroulastef@hotmail.com (D.T.S.); mtheochari@gmail.com (M.T.); 2GeneKor Medical S.A., 15344 Gerakas, Greece; eirinipapad@genekor.com (E.P.); bmetaxa@genekor.com (V.M.-M.); g.kapetsis@genekor.com (G.K.); a.meintani@genekor.com (A.M.); gtsaousis@genekor.com (G.N.T.); gnasioulas@genekor.com (G.N.); 3First Department of Pathology, School of Medicine, National Kapodistrian University of Athens, 11527 Athens, Greece; sakellarioustrat@yahoo.gr; 4First Department of Surgery, Laikon General Hospital, School of Medicine, National Kapodistrian University of Athens, 11527 Athens, Greece; felek@med.uoa.gr; 5Department of Medical Oncology, Bioclinic Hospital, 54622 Thessaloniki, Greece; ibouk@otenet.gr; 6Oncology Department, General Hospital of Rhodes, 85133 Rhodes, Greece; dimalexpoli@yahoo.com; 7Hematology Oncology Unit, Fourth Department of Internal Medicine, School of Medicine, National Kapodistrian University of Athens, 11527 Athens, Greece; akoumari@yahoo.com; 8Oncology Department, St. Lukes Hospital, 55236 Thessaloniki, Greece; zairi.eleni@gmail.com; 9Department of Clinical Therapeutics, Alexandra Hospital, School of Medicine, National Kapodistrian University of Athens, 11527 Athens, Greece; mliontos@gmail.com (M.L.); koutsoukos.k@gmail.com (K.K.)

**Keywords:** pancreatic ductal adenocarcinoma, genomic mutation, next generation sequencing, KRAS, HRR

## Abstract

**Simple Summary:**

Even in the era of precision medicine, the genomic background of pancreatic ductal adenocarcinoma has not yet been fully elucidated in large-scale populations all over the world, including Europe. The genomic characteristics from 409 Greek/South European patients with PDAC were detected by panel-based NGS and confirmed recurrent somatic alterations in *KRAS* (81.20%), *TP53* (50.75%), *CDKN2B* (8%) *SMAD4* (7.50%), and *BRCA1/2* variants (2%), among others. The majority of HRR-alterations were in intermediate- and low-risk genes (*CHEK2*, *RAD50*, *RAD51*, *ATM*, *FANCA*, *FANCL*, *FANCC*, *BAP1*), with controversial actionability. Elevated genomic LOH (gLOH) was associated with HRR-mutated status and TP53 mutations, while the lowered gLOH was associated with KRAS alterations. The comprehensive knowledge of NGS status, including TMB, MSI, and PD-L1, increased the possibility of immunotherapy use from 1.91% to 13.74%. TMB was slightly increased in females and in elderly individuals. PD-L1 > 1% either in tumor or immune cells was detected in 28.41%, PD-L1 ≥ 10% in 15.75%, PD-L1 ≥ 50% in 1.18% of cases. This is the largest NGS depiction of real-world genomic characteristics of South European patients with PDAC, which offers some useful clinical and research insights, describing the incidence of potentially targetable and predictive biomarkers and identifying genetic subtypes more susceptible to respond to specific treatments.

**Abstract:**

Despite ongoing oncological advances, pancreatic ductal adenocarcinoma (PDAC) continues to have an extremely poor prognosis with limited targeted and immunotherapeutic options. Its genomic background has not been fully characterized yet in large-scale populations all over the world. Methods: Replicating a recent study from China, we collected tissue samples from consecutive Greek patients with pathologically-confirmed metastatic/unresectable PDAC and retrospectively investigated their genomic landscape using next generation sequencing (NGS). Findings: From a cohort of 409 patients, NGS analysis was successfully achieved in 400 cases (56.50% males, median age: 61.8 years). Consistent with a previous study, *KRAS* was the most frequently mutated gene in 81.50% of tested samples, followed by *TP53* (50.75%), *CDKN2* (8%), and *SMAD4* (7.50%). *BRCA1/2* variants with on-label indications were detected in 2%, and 87.50% carried a variant associated with off-label treatment (*KRAS*, *ERBB2*, *STK11*, or HRR-genes), while 3.5% of the alterations had unknown/preliminary-studied actionability (*TP53/CDKN2A*). Most of HRR-alterations were in intermediate- and low-risk genes (*CHEK2*, *RAD50*, *RAD51*, *ATM*, *FANCA*, *FANCL*, *FANCC*, *BAP1*), with controversial actionability: 8% harbored a somatic non-*BRCA1/2* alteration, 6 cases had a high-risk alteration (*PALB2*, *RAD51C*), and one co-presented a *PALB2/BRCA2* alteration. Elevated LOH was associated with HRR-mutated status and *TP53* mutations while lowered LOH was associated with *KRAS* alterations. Including TMB/MSI data, the potential benefit from an NGS-oriented treatment was increased from 1.91% to 13.74% (high-MSI: 0.3%, TMB > 10 muts/MB: 12.78%). TMB was slightly increased in females (4.75 vs. 4.46 muts/MB) and in individuals with age > 60 (4.77 vs. 4.40 muts/MB). About 28.41% showed PD-L1 > 1% either in tumor or immune cells, 15.75% expressed PD-L1 ≥ 10%, and only 1.18% had PD-L1 ≥ 50%. This is the largest depiction of real-world genomic characteristics of European patients with PDAC, which offers some useful clinical and research insights.

## 1. Introduction

Pancreatic cancer is a fatal malignancy, primarily because it is generally diagnosed at an advanced stage, with a 5-year overall survival (OS) rate of 3–14% for metastatic or unresectable disease [1]. The most prevalent histological type is pancreatic ductal adenocarcinoma (PDAC), while the main treatment option either in an adjuvant or metastatic setting remains conventional chemotherapy, with limited novel agents being added during the last decades [2]. Moreover, even among cases with histologically proven PDAC, the inter- and intratumor heterogeneity has led to substantial disparities in treatment response and OS rates [3]. Recent bioinformatic technologies and high-throughput sequencing platforms, such as those of The Cancer Genome Atlas (TCGA) and the International Cancer Genome Consortium (ICGC), have given the opportunity for a deeper understanding of the complex molecular landscape of many cancer subtypes, including PDAC, and paved the way for precision medicine [4,5,6]. 

Despite the increasing number of targeted agents approved to treat cancers harboring specific molecular biomarkers, there is a lack of clarity as to when a tumor genomic profiling should be ordered, what type of sequencing assays should be performed, and how to interpret the results for making a treatment decision. According to ESMO Precision Medicine Working Group, it is not recommended to perform next generation sequencing (NGS) in patients with advanced PDAC in daily practice [7]. Considering the unmet medical needs and the high number of alterations ranked as level II-IV, ESMO considers that multigene sequencing could be proposed to patients with advanced PDAC in the context of molecular screening programmes to get access to innovative drugs. If NGS is not carried out, detection of druggable alterations such as microsatellite instability (MSI) status and neurotrophic tyrosine receptor kinase (*NTRK*) fusions should be done using cheaper standard methods [7]. An ASCO provisional clinical opinion tried to address the appropriate use of tumor genomic sequencing in patients with metastatic or advanced solid tumors. This ASCO report suggested that multigene panel-based assays should be performed by certified laboratories, if there are one or more specific predictive alterations that have certain regulatory-approved biomarker-linked therapeutic options. Site-agnostic approvals of agents for solid tumors with a high TMB, high MSI, or *NTRK* fusions provide a further rationale for this indication. When few or no NGS-based therapies are available for the patient’s disease, multigene testing may also assist to build a therapeutic algorithm by giving additional predictive and prognostic biomarkers. In order to make a treatment decision, the clinician should consider the functional impact of genomic alterations and the expected efficacy of targeted therapies compared to other approved or investigational options [8]. A recent study showed that there is a substantial survival benefit in PDAC patients receiving genomically guided treatment compared to those receiving conventional chemotherapy (2.58 vs. 1.51 years) [9,10].

However, there are barriers to the implementation of precision medicine in PDAC that include the heterogeneous and low individual frequencies of most actionable changes across the population, the difficulties in accessing and sequencing high-quality biopsied samples in a timely fashion, and the natural propensity of disease for rapid clinical decline [11,12,13]. In fact, approximately 25% of PDAC contain actionable molecular alterations, which are defined as alterations in driver genes (*KRAS*, *TP53*, *CDKN2A*, and *SMAD4*) that have a significant association with PDAC patients’ outcome [13,14]. Other studies have demonstrated that germline mutation carriers in homologous recombination repair (HRR) genes with PDAC have a significantly longer OS than non-carriers [15,16]. However, the genomic information may not only have a prognostic but also a predictive clinical value. Beginning from the site-agnostic indication of pembrolizumab for patients with MSI unstable tumors [17] and the TRK inhibition of NTRK fusion-positive cancers [18,19,20], there is an ongoing effort to identify further genomic- or immune-mediated biomarkers that can predict the response to certain treatments independently of tumor histology. The *KRAS* gene was untouchable for decades, being mutated in the majority of patients with PDAC. Recently, two inhibitors (Sotorasib and Adagrasib) were approved by the US Food and Drug Administration (FDA) for the treatment of previously-treated patients with *KRAS* G12C-mutated NSCLC and are under clinical testing for many other *KRAS* G12C-mutated tumor types [21]. The efficacy of both agents has also been observed in patients with advanced *KRAS* G12C-mutated PDAC, but this mutation is very rare in PDAC and only a minor proportion of patients would benefit from G12C-targeted therapy. Several compounds against the most common *KRAS* alteration, G12D, are now under development [22]. For PDAC patients with *BRCA1*/2 germline alterations, FDA has approved inhibitors of poly(ADP-ribose) polymerases (PARPs). Beyond *BRCA1*/*2*, several other genes involved in HRR pathway can cause certain genomic scars that increase the sensitivity to PARP inhibition (PARPi) and platinum-based chemotherapy [23,24]. Moreover, *BRAF* mutations and *RET* fusions are gradually gaining a tumor-agnostic FDA approval, while many other site-agnostic targets (e.g., *ERBB2*) are under evaluation in clinical trials. Even though the low frequency of abovementioned individual alterations can be detected separately, in total they may also have an impact in the tumor’s immune sensitivity [25]. To quantify this effect, Tumor Mutational Burden (TMB) was defined as the number of somatic mutations in coding regions per megabase (muts/Mb) of examined genome. The clinical utility of TMB as a predictive biomarker for anti-PD1 immunotherapy was established in the KEYNOTE-158 trial, which led to the site-agnostic FDA approval of pembrolizumab for metastatic/untreatable solid tumors with tissue TMB value ≥ 10 muts/MB [26,27]. Finally, elevated genome-wide loss of heterozygosity (gLOH) also showed a strong correlation with biallelic alterations in a core set of HRR-associated genes, such as *BARD1*, *PALB2*, *FANCC*, *RAD51C*, and *RAD51D* in breast, ovarian, pancreatic, and prostate cancer, offering further insights for examining PARPi in these patients [28].

The known genomic landscape of PDAC patients is mainly based on biomarker analyses of phase III clinical trials or based on small cohorts. Conducted a few months after a large-scale description of genetic characteristics of unresectable/metastatic PDAC in the Chinese population [14], the present study aims to provide a comprehensive depiction of the genomic profile of PDAC, using a panel-based NGS in a large South European population. The incidence of specific mutations, the percentage of targetable alterations, and the existence of immunotherapeutic biomarkers (e.g., TMB, MSI, PD-L1) were thoroughly explored in order to better understand the molecular tumor subtypes and to recognize the real-world likelihood of incorporating newer agents in the management of PDAC. 

## 2. Methods

### 2.1. Patient Samples

All consecutive patients with unresectable or metastatic PDAC who were referred by their medical oncologists to Genekor’s laboratory for multigene NGS panel-based profiling from November 2017 to April 2023 were included in our study. Due to the lack of indoor NGS infrastructure, deficiencies in the reimbursement process, and its exception from national patients’ insurance, NGS analysis is not performed in Greek public oncology departments and PDAC patients cover the cost of this genetic test. Genekor is the largest private laboratory in Greece, collecting tissue/blood samples for genetic testing from the entire country, including private clinics and community- and university-affiliated centers. In patients that relapsed after pancreatectomy (e.g., Whipple procedure), the histological tissue from the original surgery was used. In cases that were initially diagnosed at metastatic or unresectable setting without a prior operation, a tissue sample of primary site, liver metastasis or lymph node involvement was received by endoscopic ultrasound or CT-guided biopsy before first-line/neoadjuvant chemotherapy. All eligible cases had a pathologically-confirmed diagnosis of PDAC. Every analysis was performed using the most recent formalin-fixed, paraffin-embedded (FFPE) tissue specimen available before any treatment, from the primary or metastatic site. Information concerning patient (e.g., sex, age, etc.) and PDAC characteristics (e.g., tumor metastatic load, etc.) were recorded. Prior to reporting of their results, all participating patients signed the standard written informed consent for NGS analysis provided by Genekor’s laboratory that was also approved by the Medical Ethics Committee of university-affiliated Laiko General Hospital, Athens, Greece. 

### 2.2. Tissue Selection and Nucleic Acid Isolation

Genomic DNA and RNA were isolated from FFPE tumor biopsies using the MagMAX™ Total Nucleic Acid Isolation Kit (Thermo Fischer Scientific, Waltham, MA, USA) according to the manufacturer’s instructions. Nucleic acid isolation was conducted in the areas of the FFPE block with the majority of tumor cell content (TCC), as indicated by experienced pathologists in hematoxylin- and eosin-stained sections. The minimum required TCC was >20%, in a tumor area of >4 mm^2^.

### 2.3. Next Generation Sequencing (NGS)—Tumor Mutational Burden (TMB) and Microsatellite Instability (MSI) Analysis

Tumor molecular profile analysis was performed using the Oncomine Comprehensive Assay v3 (OCAv3) or Oncomine Comprehensive Assay plus (OCAplus) (Thermo Fischer Scientific, Waltham, MA, USA), which are amplicon-based targeted NGS assays, analyzing 161 and 513 unique genes respectively. The genes contained in both NGS panels are listed (513-gene panel and 161-gene panel) in Appendix A. These panels allow the identification of various mutation types such as Single Nucleotide Variants (SNVs), insertion-deletions (ins/dels), Copy Number Variations (CNVs), and gene fusions. Sequencing data were aligned against the human reference assembly GRCh37/hg19. Run metrics were accessed in the Torrent Suite™ software (version 5.18.1), using the coverage analysis plugin v5.0.4.0. NGS data analysis was completed with the Ion Reporter 5.18.4.0 software (Thermo Fisher Scientific, Waltham, MA, USA) using the manufacturer’s provided workflows. Furthermore, the analysis software Sequence Pilot (version 4.3.0, JSI medical systems, Ettenheim, Germany) was used for variant annotation. In addition to the NGS examination for identifying targetable mutations/alterations, further analysis for immunotherapeutic biomarkers such as TMB, MSI, and PD-L1 expression was also requested in each case by treating oncologists but was performed whenever the patients agreed and financially covered the cost of both tests. TMB and MSI analysis was carried out using the OCAplus Assay (Thermo Fischer Scientific, Waltham, MA, USA). This assay was also used to measure genomic instability by calculating the percentage of sample-level gLOH in addition to the analysis for HRR gene alterations. The analysis was performed using the appropriate workflow (Oncomine Comprehensive Plus—w2.5—DNA—Single Sample) in the Ion Reporter Software. 

### 2.4. Classification of Variants

Variants were classified according to their predictive value using the four-tiered system jointly recommended by the Association for Molecular Pathology (AMP), the American College of Medical Genetics (ACMG), the ASCO, and the College of American Pathologists (CAP) for the classification of somatic variants [29]. Tier 1 variants have the greatest clinical significance and include biomarkers associated with sensitivity or resistance to FDA-approved treatments, predictive biomarkers proposed by professional guidelines, and biomarkers with a strong consensus regarding their predictive significance. Tier 2 includes biomarkers with potential clinical relevance related to off-label or investigational treatments that can be used as an inclusion criterion for patient enrollment in clinical trials, as well as variants that have demonstrated predictive value in preclinical studies. Tiers 3 and 4 include biomarkers with unknown clinical significance and benign/likely benign biomarkers, respectively (Appendix A).

### 2.5. PD-L1 Expression by Immunohistochemistry

The level of PD-L1 expression was defined as the percentage of viable tumor cells (TC) showing partial or complete membrane staining at any intensity, and the percentage of tumor-infiltrating immune cells (IC) showing staining at any intensity was also calculated [30,31,32]. The analysis was conducted using the Immunohistochemistry (IHC) VENTANA PD-L1 (SP263) Assay (Roche Diagnostic, Rotkreuz, Switzerland) that utilizes the Monoclonal Mouse Anti-PD-L1, Clone SP263, accompanied by OptiView DAB IHC Detection Kit on a VENTANA BenchMark Series automated staining instrument. 

### 2.6. Statistical Analysis

Two-sided Fisher’s exact test was used to compare the median TMB values and the percentages of TMB positivity of selected groups of patients (male/female, aged > 60 y/aged < 60 y) with SPSS (version 20. IBM SPSS STATISTICS). The *p*-values were based on two-sided Fisher’s exact test. A *p*-value < 0.05 was considered to be statistically significant. Box plots were created using the Plotly.js charting library. Pathway enrichment analysis was performed against KEGG pathways using Enrichr [33].

## 3. Results

### 3.1. Characteristics of Study Population

In our study, tumor tissues from the primary tumor (pancreas) or from distant metastases (from the liver, peritoneum, lymph nodes, lung, and other sites) were obtained from a total of 409 patients with pathologically confirmed PDAC (Table 1). Briefly, 56.50% (226/400) of the patients were male, and the median age at the time of diagnosis for both sexes was 61.80 years. Compared to the general population of patients with metastatic/unresectable PDAC (median age at the time of diagnosis: 70 years), the median age of our cohort was lower, indicating a point of selection bias probably induced by the greater awareness for utilizing updated precision tools by a younger patient population [34]. The median TMB level was 4.78 (0–45.21) in 313 patients and it was slightly increased in females compared to males (4.75 vs. 4.46) and in patients with age of diagnosis >60 years compared to younger individuals (4.77 vs. 4.40) (Figure 1).

### 3.2. Genomic Profile of Somatic Alterations in Patients with Metastatic/Unresectable PDAC

Successful molecular analysis was achieved in 400 of the 409 patients analyzed, while in nine cases (2.20%) no results could be obtained due to low DNA quality or quantity (Figure 2). In total, 873 mutations were identified in 145 genes by NGS-based panels. Of the variants detected, 92.1% consisted of SNV/small indels, while 6.64% were CNVs and 1.37% were fusions. One alteration was detected in 26.75%, 2 in 33.00% and ≥3 in 32.00% of PDAC patient samples, and at least one alteration was identified in 370 cases (92.50%). Consistent with previous studies, *KRAS* was identified as the most frequently altered gene in 81.50% of tested samples, followed by *TP53*, *CDKN2A*, and *SMAD4* (50.75%, 8.00%, and 7.50% mutation frequency, respectively) (Figure 3, Appendix A) [14]. The G12D alteration was the most frequently observed *KRAS* mutation, accounting for 36.50% of tumor samples, while the G12V alteration was detected in 21.75% of analyzed tumor samples. Additionally, in four cases (1%), the G12C *KRAS* alteration, which can be targeted by emerging therapeutic agents, was identified. There was no clear correlation between *KRAS* mutations and patient age (*p* = 0.116 (two-sided Fisher’s exact test)) in the entire group; however, a subgroup analysis of somatic mutations in individuals younger than 50 years revealed a marginally lower prevalence of KRAS mutations compared to those older than 50 years (72.13% in <50 years vs. 83.38% in ≥50 years, *p* = 0.0369). Moreover, these mutations were more common in female patients (*p* = 0.019 (two-sided Fisher’s exact test) (Table 2). We found that patients with *KRAS* mutations were significantly associated with mutations in other driver genes, namely, *TP53*, *SMAD4*, and *CDKN2A* (56.31% versus 43.69%, 8.92% versus 91.08%, and 9.54% versus 90.46%, respectively). The majority of genomic alterations were clustered in the PI3K-Akt signaling, Cell Cycle, and FoxO signaling pathways.

### 3.3. Distribution of HRR Gene Alterations in Patients with Metastatic/Unresectable PDAC

As a part of our analysis, we evaluated somatic mutations in 19 HRR genes covered by our NGS panels. We found that 40 (10%) PDAC patients were accompanied by HRR gene mutations. The distribution of HRR mutant genes is displayed in Figure 4. More specifically, *BRCA1*/2 genes were altered in 2.00% of tested tumors and 8.00% of tested cases harbored a non-*BRCA1*/2 alteration with more frequent *ATM* (1.50%) and *PALB2* (1.50%). A high-risk gene alteration was present in only six patients (*PALB2* and *RAD51C*), while in one case a *PALB2* alteration was detected simultaneously with a *BRCA2* and an *ATM* alteration. Out of 13 patients, 11 with high-risk HRR alterations were male, and the mean age at the time of diagnosis was 64.54 years. The majority of HRR alterations were in an intermediate- or low-risk gene (*CHEK2*, *RAD50*, *RAD51*, *ATM*, *FANCA*, *FANCL*, *FANCC*, *BAP1*, BARD1, NBN), with controversial actionability (Figure 4). In 79 cases gLOH was also available. In agreement with a previous report [28], elevated gLOH was observed in 10 of 14 patients harboring an HRR alteration (71.43%), compared to only 21 of 65 patients without a tumor HRR alteration (32.31%). High gLOH was associated with the presence of a HRR gene mutation (*p* < 0.01 (two-sided Fisher’s exact test)). There was also a slight correlation between the presence of *TP53* mutations and elevated gLOH (*p* = 0.0369 (two-sided Fisher’s exact test)). On the contrary, the presence of a *KRAS* alteration was related to a lower gLOH value (*p* < 0.01 (two-sided Fisher’s exact test)).

### 3.4. Analysis of Immunotherapeutic Biomarkers in PDAC—Incorporation of NGS Information in Clinical Decision

Analysis of immunotherapeutic biomarkers such as MSI and TMB was conducted in 313 cases. Only 1 of 313 (0.3%) patients had a confirmed MSI-high status, while a TMB value > 10 muts/MB was detected in 40 cases (12.78%). Among the 254 patients with available PD-L1 expression, 75 (28.41%) showed a PD-L1 value more than of 1% either in tumor (TC) or immune cells (IC). Of those, 40 cases (15.75%) exhibited a more intense PD-L1 expression with a TC or IC value ≥ 10%, and these values were ≥ 50% in only three (1.18%) tumors.

Re-considering the potential of druggable alterations, at least one variant with on-label indication was identified in 2.00% of cases due to *BRCA1*/2 alterations and 87.50% carried a variant in a gene associated with off-label treatment (*KRAS G12C, ERBB2*, *STK11*, HRR-related genes and others), while 3.5% of the variants were with unknown actionability or associated with a biomarker investigated in early clinical trials (mainly *TP53* and *CDKN2A* alterations). In cases where both immunotherapeutic and genomic biomarkers were evaluated, the addition of TMB/MSI/PD-L1 analysis increased the rate of patients with an approved indication based only on NGS profiling (without TMB/MSI/PD-L1 information) from 1.91% to 13.74% (including TMB/MSI/PD-1 information). 

## 4. Discussion

This study gives a real-world genomic depiction of unresectable/metastatic PDAC based on a large-scale population from South Europe. We collected tissue samples from Greek patients with pathologically-confirmed metastatic/unresectable PDAC, retrospectively examined their genomic landscape using NGS-based gene panels, and, in parallel comparison with the recent results from Chinese population, critically debated our findings. With a failure rate of 2.2%, the used approach was successful in obtaining accurate results in 97.8% of analyzed cases, highlighting the significance of employing appropriate techniques even for low-quality DNA, such as that obtained from FFPE tissues. These results are comparable to the findings of previous studies employing the same platform in various histological types [35,36,37]. 

In 92.50% of tumor specimens analyzed, at least one alteration was identified. In agreement with Zhang et al., the oncogenic alterations in *KRAS* and *TP53* (detected in 83.00% and 51.75% of the analyzed samples, respectively) were the major molecular events in PDAC patients. Mutation of *KRAS* gene leads to permanent activation of the respective protein kinase which acts as a genetic switch to various cellular signaling pathways and transcription factors, inducing proliferation, invasion, migration, and survival [38]. In our Greek cohort, G12D was the most prevalent *KRAS*-activating variant, accounting for 36.50%, which was consistent with the mutation frequency reported in previous studies [14,38]. We found that *KRAS* mutations were more likely to occur in female patients with no association with age, in contrast to previous reports where *KRAS* mutations were more common in older patients [39,40]. *KRAS* mutation was significantly associated with three tumor suppressor genes, *TP53*, *CDKN2A*, and *SMAD4*, genes describing specific molecular subtypes, and according to Qian et al., patients who accumulated a greater number of altered driver genes had worse DFS and OS [13]. Recently, Pan et al. found that worse prognosis of *KRAS* mutation versus wild-type *KRAS* was primarily driven by the subgroup of patients who also bore *CDKN2A* mutation [41]. It has already been shown that *KRAS* G12D-mutated patients with PDAC have a significantly shorter OS compared to patients with other variants, including G12V, G12R, or wild-type patients [40].

Regarding the second most altered gene, *TP53* remains one of the four main driver genes for PDAC, including *KRAS*, *CDKN2A*, and *SMAD4* [42]. The spectrum of p53 mutations (mu*TP53*) is extremely broad, with approximately 350 alterations identified across malignancies, including deletions, missense mutations, nonsense mutations, frame shifts, etc. [43,44], and until recently, studies investigating the role of *TP53* mutations in the prognosis of PDAC have lumped all alterations together. However, the biology of each *TP53* mutation seems to be likely more complex than this described setting [45,46]. Different *TP53* mutations possess different biologic properties; some have gain-of-function properties, whereas others drive to loss-of-function [41]. Last year, Pan et al. showed that *TP53* gain-of-function mutations were associated with worse prognosis compared with *TP53* non-gain-of-function mutations in de novo metastatic, locally advanced, and recurrent PDAC, as well as molecular subgroups that retained wild-type or carried mutant *KRAS*, *CDKN2A*, or *SMAD4* [41]. 

In our study, the multigene NGS-panel has integrated and analyzed somatic and germline mutations in 19 HRR-related genes. Pathogenic mutations in *BRCA1* or *BRCA2* genes were detected in 2% and alterations in other HRR-related genes in 8% of evaluated tumor samples. These frequencies are significantly lower than those observed in the Chinese/Asian population, indicating a smaller contribution of HRR alterations in PDCA pathology in the Greek/European ancestry. The recently described survival benefit in patients harboring *BRCA* mutations treated with maintenance olaparib after platinum-based chemotherapy demonstrates the importance of identifying targetable molecular phenotypes in unresectable/metastatic PDAC [24]. However, further studies are required to determine the benefit of PARPi outside of the maintenance setting, and targeting of altered HRR genes other than *BRCA1*/2 should be evaluated in a wide range of cancer subtypes beyond PDAC. For instance, *PALB2* and *RAD51C* genes are also HRR genes with a sustained association to PARPi [47]. A recent study showed an association between germline homologous recombination deficiency status (associated with pathogenic germline *BRCA1*, *BRCA2*, *RAD51*, and *ATM*) and sensitivity to nivolumab/ipilimumab combination, advancing previous evidence of an association between *BRCA1/2* variants in other tumors and immunotherapy response [48]. On the other side of their promising inhibition, most HRR gene alterations are expressed in significantly reduced frequencies, making it difficult to determine their predictive value. However, HRR gene mutations have also been linked to the presence of gLOH, an additional biomarker of PARPi sensitivity. Several tumor types, particularly breast, ovarian, pancreatic, and prostate cancer, exhibited a strong correlation between elevated gLOH and biallelic alterations in multiple key HRR genes beyond *BRCA1* and *BRCA2*. In contrast, monoallelic/heterozygous alterations in HRR genes were not linked to elevated gLOH [28]. Furthermore, gLOH has been associated with non-HRR gene alterations, such as *TP53* loss and *KRAS* gene alterations [28]. This was also observed in our cohort, where patients with an HRR alteration had a higher gLOH than those without an HRR alteration (*p* < 0.01). There was also a modest association between the presence of *TP53* mutations and an elevated gLOH value (*p* = 0.0369). In contrast, the presence of a *KRAS* mutation was associated with a reduced gLOH value (*p* < 0.01). Further investigation of genomic instability contribution to PARPi treatment response is required, while large studies investigating its incidence in PDAC patients were limited up until to now. To best of our knowledge, such an assessment of gLOH status is also missing in Asian population.

It is also notable that among the 313 cases analyzed for immunotherapeutic biomarkers, only 1 case (0.32%) exhibited MSI positivity, whereas 40 cases (12.78%) demonstrated a TMB value > 10 muts/MB. Usually high-TMB appeared as a rarer but not-negligible molecular feature, being present in about 1.1% of cases [49]. TMB-high cases usually belong to specific PDAC subsets with prolonged survival, further actionable alterations, and also high MSI status, displaying strong anti-tumor cytotoxic T-cell-mediated immune response [50]. Elevated TMB levels have been retrospectively correlated with response to immune checkpoint inhibition in different cancer subtypes and recently the phase II KEYNOTE-158 trial in ten tumor-type-specific cohorts prospectively identified that a subgroup of patients with high tissue TMB status could have a robust tumor response to pembrolizumab monotherapy [51]. This study supported the tissue agnostic indication of pembrolizumab for the treatment of untreatable/metastatic solid tumors with a TMB value of ≥10 muts/MB and heightened the interest in screening this biomarker [27]. A recent case-report describes a profound clinical response to sequential platinum-based chemotherapy, pembrolizumab, and olaparib in a patient with metastatic PDAC harboring a germline *BRCA1* mutation and extremely high TMB. Noticeably in this case, considering the immune sensitivity of his tumor despite microsatellite stability, the patient elected to self-fund pembrolizumab in addition to platinum-based chemotherapy, reaching near complete resolution of the primary and metastatic sites [11]. Even among the general low TMB values in pancreatic cancer setting, a further stratification after whole-exome sequencing and gene expression profiling of 93 resected pancreatic cancer cases showed that those with TMB-ultra-low (<1 muts/MB) had significantly fewer borderline resectable lesions, had fewer adeno-squamous histologies, showed significantly lower detection rates of driver mutations and copy number variations, and had a significantly better prognosis than others with TBM-low (<5, ≥1 muts/MB) [52]. The sample size of high-TMB PDAC patients treated with ICIs so far globally is quite small [49] but if the information of PD-L1 expression is added (28.41% of our samples showed a PD-L1 value > 1% either in tumor or in the immune cells), there is potential space for off-label use of immunotherapy in some specific cases. This study may represent a reliable starting point for the baseline assessment of TMB and PD-L1 in patients with metastatic/unresectable PDAC in order to guide the entire clinical management.

The main limitations of our study are coming from its retrospective design. It suffers from selection bias (e.g., “selection” by patients’ themselves, two different gene panels were used, TMB/MSI/PD-L1 testing in smaller subgroups); however, the collection and the detection of all samples were performed in real-time with satisfied quality control. Treating oncologists suggested the NGS and immune biomarker analyses in each PDAC patient that they faced during the reported period in order to build up a more individualized treatment plan/sequence. However, both analyses were completed only in cases that agreed with this proposed strategy and supported it. This could be the reason for the lower median age at PDAC diagnosis in our cohort compared to the median age recorded in the overall population. In general, younger patients are more eager to support genomically-oriented precision approaches than the older ones. In addition, our NGS-based gene panels involved the majority of the vital genes in PDAC pathogenesis, but some potentially valuable genes may have been overlooked. Similar to the large recent Chinese study, therapeutic profiling and prognosis information matching genomics are also missing in our cohort. The abovementioned potential gaps, primarily in the selection process, may undermine the generalizability of our findings. Prospective well-designed clinical trials should be conducted in the close future with a more combined and comprehensive approach involving whole-genome, transcriptome, and proteome multi-omic analyses. 

## 5. Conclusions

In conclusion, this study highlights the clinical significance of screening genomic and immunotherapeutic biomarkers at baseline in real-world patients with unresectable/metastatic PDAC, describing the largest, until now, NGS depiction of South European PDAC patients. In our retrospective report, we noticed that the incorporation of this information could increase the percentage of candidates for targeted or immunotherapeutic agents to 13.74% (compared to only 1.91% in those without PD-1/TMB/MSI analysis). Our findings were closely consistent with the Chinese study but more data in a different subpopulation have been accumulated, standardizing the clinical significance of NGS analysis in PDAC patients and supporting in general the expanding capabilities of genomic sequencing. The implementation of NGS analysis into routine care of PDAC patients not only prompts the exploration of potential therapeutic targets but also updates the understanding of disease pathobiology and provides insights for a more personalized approach in PDAC management. 

## Figures and Tables

**Figure 1 cancers-16-00002-f001:**
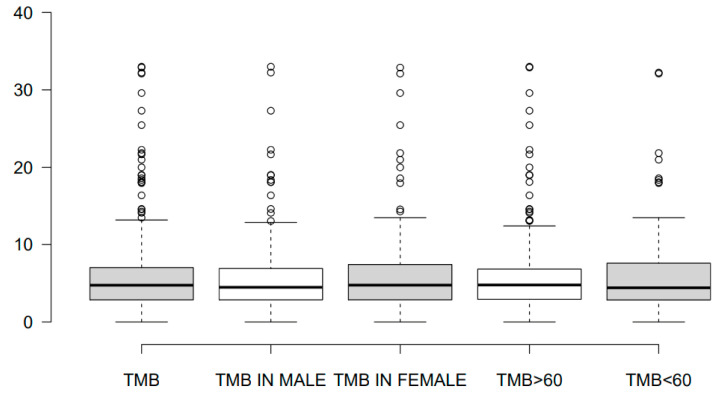
Box plots showing the median TMB values in pancreatic cancer patients and related to patients’ gender and age of disease onset (> or <60 years).

**Figure 2 cancers-16-00002-f002:**
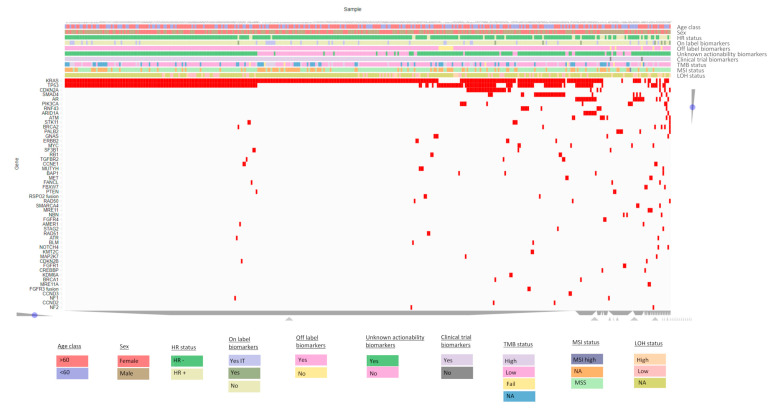
Heatmap showing patients characteristics and gene alterations found in 400 pancreatic cancer patients. An interactive form of the heatmap can be found at https://maayanlab.cloud/clustergrammer/viz/648123fab1ed870ccef99043/pancreas.data.matrix_up.txt (accessed on 1 October 2023).

**Figure 3 cancers-16-00002-f003:**
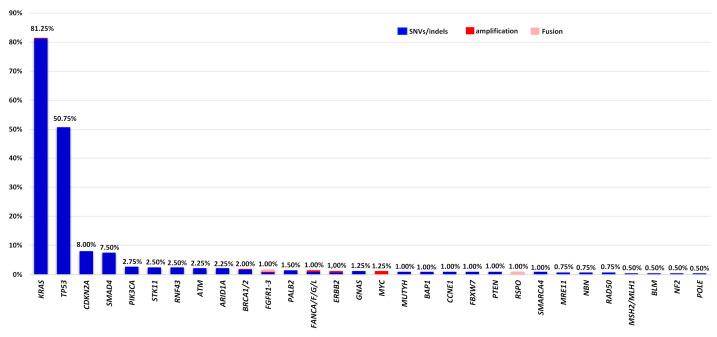
The top 30 most frequently altered genes in our cohort with PDAC patients.

**Figure 4 cancers-16-00002-f004:**
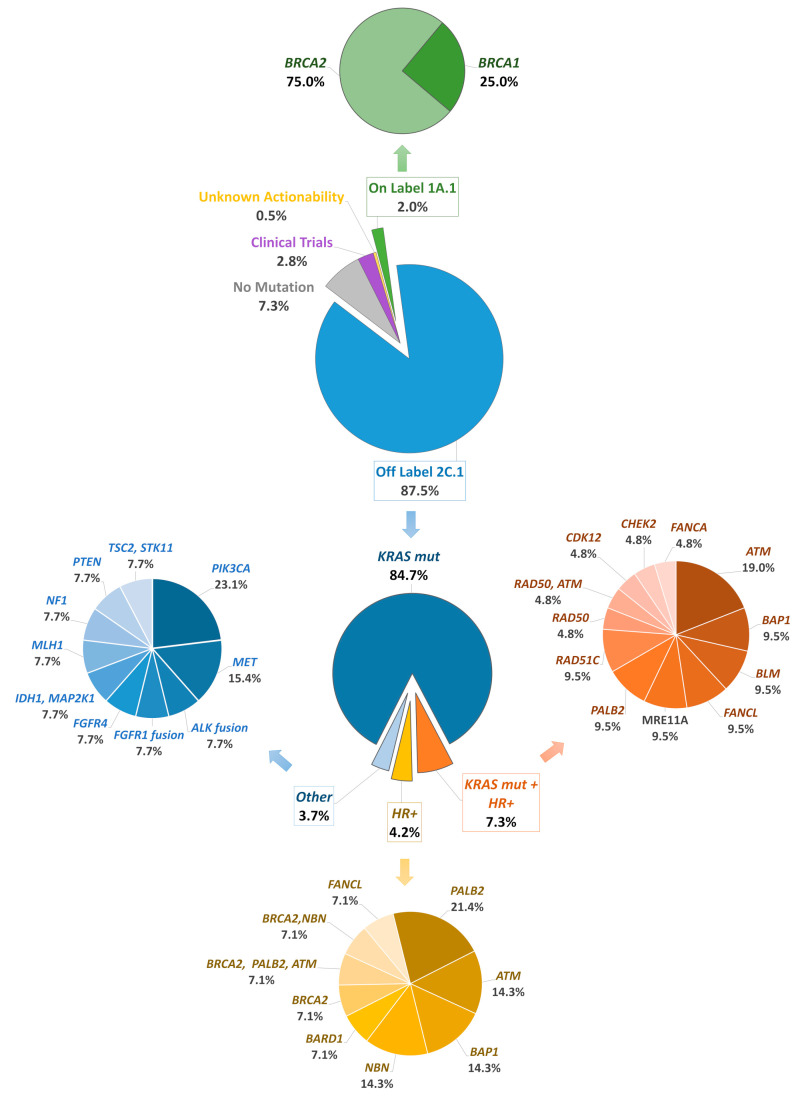
The distribution of homologous recombination repair (HRR) gene alterations in the 40 HRR positive tumors.

**Table 1 cancers-16-00002-t001:** Characteristics of the study population.

Features	No. of Patients	%
**Age**		
<60	165	41.25%
≥60	235	58.75%
**Gender**		
Male	226	56.50%
Female	174	43.50%
**TMB level**		
Median (Range)	313	4.78 (0–45.21)

**Table 2 cancers-16-00002-t002:** Biomarkers’ correlation to patients’ age and gender.

Features	Biomarker				
	TMB > 10 muts/MB	KRAS Mutation	TP53 Mutation	HR Gene Positive	%LOH > 16%
Male	11.18%	77.43%	51.72%	7.25%	24.05%
Female	14.69%	86.78%	48.28%	2.75%	15.19%
Age < 60	13.85%	77.58%	43.35%	4.00%	20.25%
Age > 60	12.02%	84.25%	56.65%	6.00%	18.99%
Total	12.78% (40/313)	81.50% (326/400)	50.75% (203/400)	10.00% (40/400)	39.24% (31/79)

## Data Availability

Data and materials can be provided upon reasonable request to the corresponding author.

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
