# Peer review of "Digging into the NGS Information from a Large-Scale South European Population with Metastatic/Unresectable Pancreatic Ductal Adenocarcinoma: A Real-World Genomic Depiction"

_cancers, 2023, doi:10.3390/cancers16010002_

Round 1
Reviewer 1 Report
Comments and Suggestions for Authors
Summary: This is an interesting paper that adds additional information regarding the scope of somatic mutations among individuals with pancreatic ductal adenocarcinoma. The authors gave comprehensive information regarding the genes affected, SNV, CNV, TMB, MSI, and PD-L1 status. However, additional information would be helpful regarding the following: (i) hospital where these samples were taken (ii) if there samples were taken from individuals who were initially treatment-naïve (iii) given the number of individuals who were < 60, would also see if they could report on a subset with early onset PDAC (ie < 50)
Individual comments below:
Introduction:
1. The introduction should ideally be shortened as it’s nearly two pages. The paragraph beginning on page 2, line 102 can be shortened and focus on what has been in development or approved.
Methods
2. Please explain where these samples were obtained: clinic sites, hospitals, and explain the type of clinic or hospital . Is this is tertiary center or community based oncology clinic? Are these patients treatment naïve or do they come to this clinic with progressing cancer? It is helpful so we can understand the context of these mutations.
3. Methods, line 194: “also performed whenever requested”
Is there a protocol in place regarding when to request MSI/PD-L1 testing? If not, this should be listed as a limitation.
Results
4. The median age of pancreatic cancer was 61.8. In the United states, the average age is closer to 68-70. Can the authors comment on this?
5. Can a subanalysis of the somatic mutations be done on individuals < 50 for early onset pancreatic cancer?
Conclusions
1. Would talk more about the pros and cons of the group from which these samples arose and how similar or different they are from a general population.
Comments on the Quality of English Language
The introduction and conclusion should be shorter.
Author Response
Reviewer 1:
Summary: This is an interesting paper that adds additional information regarding the scope of somatic mutations among individuals with pancreatic ductal adenocarcinoma. The authors gave comprehensive information regarding the genes affected, SNV, CNV, TMB, MSI, and PD-L1 status. However, additional information would be helpful regarding the following: (i) hospital where these samples were taken (ii) if there samples were taken from individuals who were initially treatment-naïve (iii) given the number of individuals who were < 60, would also see if they could report on a subset with early onset PDAC (ie < 50).
Authors’ reply: We thank the reviewer for the positive general comments. The Methods’ section is now thoroughly modified including additional information regarding: (i) the referral and sampling process (ii) the time of tissue biopsy and the previous medical procedures of included patients and (iii) the number of individuals who were < 60 years old. In addition, a subset analysis of patients with early onset PDAC (ie < 50 years) is now reported in the Results section.
“All consecutive patients with unresectable or metastatic PDAC, who were referred by their medical oncologists to Genekor’s laboratory for multigene NGS panel-based profiling from November 2017 to April 2023, were included in our study. Due to the lack of indoor NGS infrastructure, to deficiencies in the reimbursement process and to its exception from national patients’ insurance, NGS analysis is not performed in Greek public oncology departments and PDAC patients should cover the cost of this genetic test. Genekor is the largest private laboratory in Greece, collecting tissue/blood samples for genetic testing from the entire country, including private clinics and community-/university-affiliated centers. In patients that relapsed after pancreatectomy (e.g. Whipple procedure), the histological tissue from the original surgery was used. In cases that were initially diagnosed at metastatic or unresectable setting without a prior operation, a tissue sample of primary site, liver metastasis or lymph node involvement was received by endoscopic ultrasound or CT-guided biopsy, before first-line/neoadjuvant chemotherapy. All eligible cases had a pathologically-confirmed diagnosis of PDAC. Every analysis was performed using the most recent formalin-fixed, paraffin embedded (FFPE) tissue specimen available before any treatment, from the primary or metastatic site. Information concerning patient’s (e.g., sex, age, etc.) and PDAC characteristics (e.g., tumor metastatic load, etc.) were recorded. All participating patients signed, prior to reporting of their results, the standard written informed consent for NGS analysis provided by Genekor’s laboratory that was also approved by the Medical Ethics Committee of university-affiliated Laiko General Hospital, Athens, Greece.”
“There was no clear correlation between KRAS mutations and patient age (p=0.116 [two-sided Fisher’s exact test]) in the entire group, however a subgroup analysis of somatic mutations in individuals younger than 50 years revealed a marginally lower prevalence of KRAS mutations compared to those older than 50 years (72.13% in <50 years vs. 83.38% in >=50 years, p=0.0369).”
Individual comments below:
Introduction
- The introduction should ideally be shortened as it’s nearly two pages. The paragraph beginning on page 2, line 102 can be shortened and focus on what has been in development or approved.
Authors’ reply: The introduction has been shortened according to reviewer’s suggestions.
Methods
- Please explain where these samples were obtained: clinic sites, hospitals, and explain the type of clinic or hospital . Is this is tertiary center or community based oncology clinic? Are these patients treatment naïve or do they come to this clinic with progressing cancer? It is helpful so we can understand the context of these mutations.
Authors’ reply: The Methods’ section is now thoroughly modified including additional information regarding: (i) the hospitals where these samples were taken (ii) the previous medical history of included patients at the time of biopsy and (iii) the number of individuals who were < 60 years old.
- Methods, line 194: “also performed whenever requested” Is there a protocol in place regarding when to request MSI/PD-L1 testing? If not, this should be listed as a limitation.
Authors’ reply: The Methods’ section is now thoroughly modified including additional information regarding the type of performed tests.
“In addition to the NGS examination for identifying targetable mutations/alterations, further analysis for immunotherapeutic biomarkers such as TMB, MSI, and PD-L1 expression was also requested in each case by treating oncologists but was performed whenever the patients agreed and financially covered the cost of both tests. TMB and MSI analysis was carried out using the OCAplus Assay (Thermo Fischer Scientific).”
Results
- The median age of pancreatic cancer was 61.8. In the United states, the average age is closer to 68-70. Can the authors comment on this?
Authors’ reply: Now, it is clearly noted in the reporting of results as following:
“Briefly, 56.50% (226/400) of the patients were male, and the median age at the time of diagnosis for both sexes was 61.80 years. Compared to the general population of patients with metastatic/unresectable PDAC (median age at the time of diagnosis: 70 years), the median age of our cohort was lower, indicating a point of selection bias probably induced by the greater awareness for utilizing updated precision tools by younger patient population [33].”
And also discussed in the Discussion section, listing the limitations of our study:
“The main limitations of our study are coming from its retrospective design. It suffers from selection bias (e.g., “selection” by patients’ themselves, two different gene panels were used, TMB/MSI/PD-L1 testing in smaller subgroups); however, the collection and the detection of all samples were performed in real-time with satisfied quality control. Treating oncologists suggested the NGS and immune biomarker analyses in each PDAC patient that they faced during the reported period in order to build up a more individualized treatment plan/sequence. However, both analyses were completed only in cases that agreed with this proposed strategy and supported it. This could be the reason of the lower median age at PDAC diagnosis in our cohort compared to the median age recorded in the overall population. In general, younger patients are more eager to support genomically-oriented precision approaches than the older ones. In addition, our NGS-based gene panels involved the majority of the vital genes in PDAC pathogenesis, but some potentially valuable genes may have been overlooked. Similar to the large recent Chinese study, therapeutic profiling and prognosis information matching genomics are also missing in our cohort. The abovementioned potential gaps, primarily in the selection process, may undermine the generalizability of our findings. Prospective well-designed clinical trials should be conducted in the close future with a more combined and comprehensive approach involving whole-genome, transcriptome, and proteome multi-omic analyses.”
- Can a subanalysis of the somatic mutations be done on individuals < 50 for early onset pancreatic cancer?
Authors’ reply: A sub-analysis of the somatic mutations has been done on individuals < 50 for early onset pancreatic cancer:
“There was no clear correlation between KRAS mutations and patient age (p=0.116 [two-sided Fisher’s exact test]) in the entire group, however a subgroup analysis of somatic mutations in individuals younger than 50 years revealed a marginally lower prevalence of KRAS mutations compared to those older than 50 years (72.13% in <50 years vs. 83.38% in >=50 years, p=0.0369).”
Conclusions
- Would talk more about the pros and cons of the group from which these samples arose and how similar or different they are from a general population.
Authors’ reply: The Methods section describes now in more details the sampling procedure:
“All consecutive patients with unresectable or metastatic PDAC, who were referred by their medical oncologists to Genekor’s laboratory for multigene NGS panel-based profiling from November 2017 to April 2023, were included in our study. Due to the lack of indoor NGS infrastructure, to deficiencies in the reimbursement process and to its exception from national patients’ insurance, NGS analysis is not performed in Greek public oncology departments and PDAC patients should cover the cost of this genetic test. Genekor is the largest private laboratory in Greece, collecting tissue/blood samples for genetic testing from the entire country, including private clinics and community-/university-affiliated centers. In patients that relapsed after pancreatectomy (e.g. Whipple procedure), the histological tissue from the original surgery was used. In cases that were initially diagnosed at metastatic or unresectable setting without a prior operation, a tissue sample of primary site, liver metastasis or lymph node involvement was received by endoscopic ultrasound or CT-guided biopsy, before first-line/neoadjuvant chemotherapy. All eligible cases had a pathologically-confirmed diagnosis of PDAC. Every analysis was performed using the most recent formalin-fixed, paraffin embedded (FFPE) tissue specimen available before any treatment, from the primary or metastatic site. Information concerning patient’s (e.g., sex, age, etc.) and PDAC characteristics (e.g., tumor metastatic load, etc.) were recorded. All participating patients signed, prior to reporting of their results, the standard written informed consent for NGS analysis provided by Genekor’s laboratory that was also approved by the Medical Ethics Committee of university-affiliated Laiko General Hospital, Athens, Greece.”
The main difference in age at PDAC diagnosis is initially noticed in the results section:
“Briefly, 56.50% (226/400) of the patients were male, and the median age at the time of diagnosis for both sexes was 61.80 years. Compared to the general population of patients with metastatic/unresectable PDAC (median age at the time of diagnosis: 70 years), the median age of our cohort was lower, indicating a point of selection bias probably induced by the greater awareness for utilizing updated precision tools by younger patient population [33].”
and the paragraph of limitations has also been enriched as following:
“The main limitations of our study are coming from its retrospective design. It suffers from selection bias (e.g., “selection” by patients’ themselves, two different gene panels were used, TMB/MSI/PD-L1 testing in smaller subgroups); however, the collection and the detection of all samples were performed in real-time with satisfied quality control. Treating oncologists suggested the NGS and immune biomarker analyses in each PDAC patient that they faced during the reported period in order to build up a more individualized treatment plan/sequence. However, both analyses were completed only in cases that agreed with this proposed strategy and supported it. This could be the reason of the lower median age at PDAC diagnosis in our cohort compared to the median age recorded in the overall population. In general, younger patients are more eager to support genomically-oriented precision approaches than the older ones. In addition, our NGS-based gene panels involved the majority of the vital genes in PDAC pathogenesis, but some potentially valuable genes may have been overlooked. Similar to the large recent Chinese study, therapeutic profiling and prognosis information matching genomics are also missing in our cohort. The abovementioned potential gaps, primarily in the selection process, may undermine the generalizability of our findings. Prospective well-designed clinical trials should be conducted in the close future with a more combined and comprehensive approach involving whole-genome, transcriptome, and proteome multi-omic analyses.”
In Conclusion section:
“Our findings were closely consistent with the Chinese study but more data in different subpopulation have been accumulated, standardizing the clinical significance of NGS analysis in PDAC patients and supporting in general the expanding capabilities of genomic sequencing.”
Reviewer 2 Report
Comments and Suggestions for Authors
Indeed, despite ongoing advances in oncology, pancreatic ductal adenocarcinoma (PDAC) has an extremely poor prognosis with limited targeted and immunotherapy options. The authors replicated the study conducted in China only on a Greek cohort of patients with pathologically confirmed metastatic/unresectable PDAC and retrospectively examined their genomic landscape using next generation sequencing (NGS). The findings were closely consistent with the Chinese study: KRAS was the most frequently mutated gene in 81.20% of samples tested, followed by TP53 (50.75%), CDKN2 (8%) and SMAD4 (7.50%).
1. The relevance of the study is not entirely obvious: accumulation of data? expanding genomic sequencing capabilities?
2. I didn’t have enough analysis of the data obtained: how is patient survival related to the presence of certain mutations? What combinations of mutations have the most unfavorable prognosis? Have combinations of mutations, etc. been assessed at all? What is the difference between your research and the Chinese one, you need to place emphasis. In its current form, this is simply a statement of the fact of the presence of certain mutations in the study cohort.
A small note: Figures 2 have unreadable captions, this needs to be corrected.
Author Response
Reviewer 2:
Indeed, despite ongoing advances in oncology, pancreatic ductal adenocarcinoma (PDAC) has an extremely poor prognosis with limited targeted and immunotherapy options. The authors replicated the study conducted in China only on a Greek cohort of patients with pathologically confirmed metastatic/unresectable PDAC and retrospectively examined their genomic landscape using next generation sequencing (NGS). The findings were closely consistent with the Chinese study: KRAS was the most frequently mutated gene in 81.20% of samples tested, followed by TP53 (50.75%), CDKN2 (8%) and SMAD4 (7.50%).
Authors’ reply:
- The relevance of the study is not entirely obvious: accumulation of data? expanding genomic sequencing capabilities?
Authors’ reply: In the conclusion section we now notice that: “Our findings were closely consistent with the Chinese study but more data in different subpopulation is now accumulated, standardizing the clinical significance of NGS analysis in PDAC and supporting the expanding capabilities of genomic sequencing.”
- I didn’t have enough analysis of the data obtained: how is patient survival related to the presence of certain mutations? What combinations of mutations have the most unfavorable prognosis? Have combinations of mutations, etc. been assessed at all? What is the difference between your research and the Chinese one, you need to place emphasis. In its current form, this is simply a statement of the fact of the presence of certain mutations in the study cohort.
Authors’ reply: For these separate issues:
- how is patient survival related to the presence of certain mutations? What combinations of mutations have the most unfavorable prognosis?
Authors’ reply: It is not feasible, chronically and practically to have such an information in so large heterogenous cohort outside a clinical trial setting. We report this study gap now in the Discussion section: “Similar to the large recent Chinese study, therapeutic profiling and prognosis information matching genomics are also missing in our cohort.”
- Have combinations of mutations, etc. been assessed at all?
Authors’ reply: The incidence of more than one mutation is now examined: “We found that patients with KRASmutations were significantly associated with mutations in other driver genes, namely, TP53, SMAD4, and CDKN2A(56,31% versus 43,69%, 8,92% versus 91,08%, 9,54% versus 90,46% respectively). The majority of genomic alterations were clustered in the PI3K-Akt signalling, Cell Cycle and FoxO signalling pathway.”
“In agreement with previous report, elevated gLOH was observed in 10 of 14 patients harboring an HRR alteration (71.43%), compared to only 21 of 65 patients without a tumor HRR alteration (32.31%). High gLOH was associated with the presence of a HRR gene mutation (p<0.01 [two-sided Fisher’s exact test]). There was also a slight correlation between the presence of TP53 mutations and elevated gLOH (p=0.0369 [two-sided Fisher’s exact test]). On the contrary the presence of a KRAS alteration was related to a lower gLOH value (p<0.01 [two-sided Fisher’s exact test]).”
- What is the difference between your research and the Chinese one, you need to place emphasis.
Authors’ reply: Throughout the Discussion section, the two studies (our research and the Chinese one) are under debate, placing more emphasis in their findings.
For instance:
“This study gives a real-world genomic depiction of unresectable/metastatic PDAC based on a large-scale population from South Europe. We collected tissue samples from Greek patients with pathologically-confirmed metastatic/unresectable PDAC, retrospectively examined their genomic landscape using NGS-based gene panels and in parallel comparison with the recent results from Chinese population, we critically debated our findings.”
“These frequencies are significantly lower than those observed in the Chinese/Asian population, indicating a smaller contribution of HRR alterations in PDCA pathology in the Greek/European ancestry.”
“Further investigation of genomic instability contribution to PARPi treatment response is required, while large studies investigating its incidence in PDAC patients were limited up until to now. To best of our knowledge, such an assessment of gLOH status is also missing in Asian population.”
“The main limitations of our study are coming from its retrospective design. It suffers from selection bias (e.g., “selection” by patients’ themselves, two different gene panels were used, TMB/MSI/PD-L1 testing was performed in smaller subgroups); however, the collection and the detection of all samples were performed in real-time with satisfied quality control. Treating oncologists suggested the NGS and immune biomarker analyses in each PDAC patient that they faced during the reported period in order to build up a more individualized treatment plan/sequence. However, both analyses were completed only in cases that agreed with this proposed strategy and supported it financially. This is the main reason of the lower median age at PDAC diagnosis in our cohort compared to the median age recorded in the overall population. In general, younger patients are more eager to support genomically-oriented precision approaches than the older ones. In addition, our NGS-based gene panels involved the majority of the vital genes in PDAC pathogenesis, but some potentially valuable genes may have been overlooked. Similar to the large recent Chinese study, therapeutic profiling and prognosis information matching genomics are also missing in our cohort. The abovementioned potential gaps, primarily in the selection process, may undermine the generalizability of our findings.”
“Our findings were closely consistent with the Chinese study but more data in different subpopulation have been accumulated, standardizing the clinical significance of NGS analysis in PDAC patients and supporting in general the expanding capabilities of genomic sequencing.”
A small note: Figures 2 have unreadable captions, this needs to be corrected.
Authors’ reply: Figure 2 is now improved, separating into 2 figures.
Reviewer 3 Report
Comments and Suggestions for Authors
A study by Ziogas et al. investigates the genomic background of pancreatic ductal adenocarcinoma has not been fully elucidated yet in large-scale populations. The study is sufficiently well performed, however, the presentation of the results must be improved. Figures should be more legible.
Specific comments:
1. Gene names should be consistently written in italics. For example, in abstract they are written correctly but in a short summary they are not.
2. Fig. 2A cannot be read at all as it is.
Comments on the Quality of English Language
minor revision
Author Response
Reviewer 3:
A study by Ziogas et al. investigates the genomic background of pancreatic ductal adenocarcinoma has not been fully elucidated yet in large-scale populations. The study is sufficiently well performed, however, the presentation of the results must be improved. Figures should be more legible.
Authors’ reply: We thank the reviewer for the positive general comments. The presentation of the results and the figures are now improved according to the specific comments.
Specific comments:
- Gene names should be consistently written in italics. For example, in abstract they are written correctly but in a short summary they are not.
Authors’ reply: All gene names are now consistently written in italics.
- 2A cannot be read at all as it is.
Authors’ reply: Fig. 2A is now improved, separating into 2 figures.
Round 2
Reviewer 2 Report
Comments and Suggestions for Authors
I have no further comments on the article. I believe that in its present form the manuscript can be recommended for publication.
Reviewer 3 Report
Comments and Suggestions for Authors
The quality of the manuscript has improved. The comments have been adequately addressed.